# Franz Josef Land: extreme northern outpost for Arctic fishes

Natalia V. Chernova[1], Alan M. Friedlander[2,3], Alan Turchik[2] and Enric Sala[2]

[1] Zoological Institute of Russian Academy of Sciences, St. Petersburg, Russia
[2] National Geographic Society, Washington, DC, USA
[3] Fisheries Ecology Research Laboratory, University of Hawaii, Honolulu, USA

Corresponding author
Alan M. Friedlander,
alan.friedlander@hawaii.edu

## ABSTRACT

The remote Franz Josef Land (FJL) Archipelago is the most northerly land in Eurasia and its fish fauna, particularly in nearshore habitats, has been poorly studied. An interdisciplinary expedition to FJL in summer 2013 used scuba, seines, and plankton nets to comprehensively study the nearshore fish fauna of the archipelago. We present some of the first underwater images for many of these species in their natural habitats. In addition, deep water drop cameras were deployed between 32 and 392 m to document the fish fauna and their associated habitats at deeper depths. Due to its high latitude (79°–82°N), extensive ice cover, and low water temperatures (<0 °C much of the year), the fish diversity at FJL is low compared to other areas of the Barents Sea. Sixteen species of fishes from seven families were documented on the expedition, including two species previously unknown to the region. One Greenland shark, *Somniosus microcephalus* (Somniosidae), ca. 2 m in length, was recorded by drop camera near Hayes Island at 211 m, and Esipov's pout, *Gymnelus esipovi* (Zoarcidae), was collected at Wilton Island at 15 m in a kelp forest. Including the tape-body pout, *Gymnelus taeniatus*, described earlier from the sub-littoral zone of Kuhn Island, 17 fish species are now known from FJL's nearshore waters. Species endemic to the Arctic accounted for 75% of the nearshore species observed, followed by species with wider ranges. A total of 43 species from 15 families are known from FJL with the majority of the records from offshore trawl surveys between 110 and 620 m. Resident species have mainly high Arctic distributions, while transient species visit the archipelago to feed (e.g., Greenland shark), and others are brought by currents as larvae and later migrate to spawn grounds in the south (e.g., Atlantic cod *Gadus morhua*, Capelin *Mallotus villosus*, Beaked redfish *Sebastes mentella*). Another species group includes warmer-water fishes that are rare waifs (e.g., Glacier lanternfish *Benthosema glaciale*, White barracudina *Arctozenus rissoi*). The rapid warming of the Arctic will likely result in significant changes to the entire ecosystem and this study therefore serves as an important baseline for the nearshore fish assemblages in this unique and fragile region.

## INTRODUCTION

Owing to its unique biogeographic and climatic histories, the Arctic Ocean has produced a distinctive fish fauna dominated by phylogenetically young families (e.g., Zoarcidae, Stichaeidae) (*Andriashev, 1939*; *Dunbar, 1968*; *Mecklenburg, Møller & Steinke, 2010*). Older groups eliminated during the rapid cooling of the Middle Miocene were followed by younger families invading the Arctic mainly from the Pacific via the opening of the Bering Strait 3–3.5 million years ago (*Andriashev, 1939*; *Savin, 1977*; *Mecklenburg, Møller & Steinke, 2010*), while a few families have also invaded the region from the Atlantic (e.g., Gadidae, Anarhichadidae) (*Svetovidov, 1948*; *Mecklenburg, Møller & Steinke, 2010*). Of the 504 species currently comprising the Arctic ichthyofauna, those with Atlantic–Arctic ranges comprise 58% of the total richness, followed by species with Pacific–Arctic ranges (20%), while those endemic to the Arctic region account for an additional 14% (*Chernova, 2011*).

The Franz Josef Land (FJL) Archipelago is located within the Barents Sea Large Marine Ecosystem, which is a transition zone where relatively warm, more saline water from the Atlantic mixes with Arctic and Polar waters (*Johannesen et al., 2012a*). These oceanographic conditions result in high productivity (*Cochrane et al., 2009*) that supports major fisheries, including the largest remaining stock of Atlantic cod (*Gadus morhua*) (*Johannesen et al., 2012b*). Currently >200 fish species from 66 families are found in the Barents Sea (*Stiansen & Filin, 2008*; *Dolgov, 2011*). The dominant families are: eelpouts (Zoarcidae), sculpins (Cottidae), codfishes (Gadidae), snailfishes (Liparidae), flatfishes (Pleuronectidae), which collectively account for nearly 80% of the species regularly occurring in the Barents Sea.

FJL is a *zakaznik* (protected area, equivalent to IUCN category IV), currently managed by the Russian Arctic National Park. Its remoteness and harsh physical environment makes it one of the least known places on earth. The archipelago is situated in the NE Barents Sea (79°–82°N, 43°–67°E) and consists of 192 islands, covering 16,134 km$^2$ (*Barr, 1994*) Although FJL lies at the same latitude as Svalbard, Norway, the fish assemblages are very different (*Fossheim, Nilssen & Aschan, 2006*; *Stiansen et al., 2009*). The warm Atlantic currents that run along the northern continental slope of the Barents Sea, are greatly diminished by the time they reach FJL (*Schauer et al., 2002*), resulting in a biota consisting mainly of cold-water organisms (*Wassmann et al., 2006*). Because of remoteness and severe ice conditions most of the year, the FJL fish fauna has not been well studied.

### Previous fish survey of FJL

The first records of fishes from FJL come from the Austro-Hungarian expedition of 1872–74, which noted blackbelly snailfish (as *Liparis gelatinosus*) and polar cod (as *Gadus morue*) (*Payer, 1878*). During the Norwegian Polar Expedition of 1893–1896, polar cod (*Boreogadus saida*) were observed in the stomachs of several sea bird species (*Collett & Nansen, 1899*). The 1894–97 British Expedition to FJL spent a winter at Cape Flora on Northbrook Island and collected a few polar cod (*Jackson, 1899*). The same species was collected by the Italian expedition led by Duke of the Abruzzi, Luigi Amedeo, who wintered on Rudolf Island in 1899–1900 (*Camerano, 1903*).

In 1901, a Russian expedition to FJL on the ice-breaker *Ermak* identified nine species of fishes at depths from 24 to 358 m. The Russian ice-breaker *Georgiy Sedov* visited FJL in 1929 and collected Polar cod, twohorn sculpin (*Icelus bicornis*), and kelp snailfish (*Liparis tunicatus*) (*Esipov, 1931*; *Esipov, 1933*). An oceanographic expedition in 1955 conducted by the Soviet Arctic Institute (now AARI, St. Petersburg) surveyed the margins of the continental shelf north of FJL using trawls and collected ten species of fishes (*Andriashev, 1964b*; *Koltun, 1964*). In 1980–90s the Knipovitch Polar Research Institute of Marine Fisheries and Oceanography (PINRO, Murmansk) conducted benthic and pelagic trawls in the north-eastern Barents Sea, mainly 50–60 nm south-west of FJL in the Franz-Victoria Trough (depth 170–620 m), and collected 33 fish species (*Borkin, 1993*).

The expedition of the Zoological Institute of the Russian Academy of Sciences (ZIN) to FJL in 1981–82 used scuba to collect nearshore fishes. In 1991 and 1992, the expedition of the Murmansk Marine Biological Institute (MMBI) conducted additional scuba sampling at FJL, resulting in new information on three species of snailfishes (*Chernova, 1993*; *Chernova, 2007*). In addition, a new zoarcid fish, the tape-body pout *Gymnelus taeniatus* was described from one specimen found in the sub-littoral zone off Kuhn Island (*Chernova, 1999a*).

Surveys of the Barents Sea were conducted from 2004 to 2009 under the Russian-Norwegian Cooperation Program, and resulted in an atlas of Barents-Sea fishes (*Wienerroither et al., 2011*). More than 30 species were listed from FJL, including the White barracudina, *Arctozenus risso*, and Esmark's eelpout, *Lycodes esmarkii*. PINRO produced a key to the identification of Barents Sea fishes that listed 11 species for FJL including the thorny ray, *Amblyraja radiata*, which was collected near Alexandra Land and Prince George Land (*Dolgov, 2011*).

Currently, a total of 43 fish species from 15 families and 9 orders are known from FJL, with most species collected by trawling southwest of the archipelago at depths from 100 to 600 m (Table S1). In the summer of 2013, an interdisciplinary research expedition to FJL, led by the National Geographic Society and the Russian Arctic National Park, conducted an assessment of the biodiversity of the ichthyofauna using a suite of sampling methods with the objectives of describing the taxonomy and ecology of the nearshore ichthyofauna, while also exploring the deep sea environment around the archipelago.

## MATERIALS AND METHODS

### Sample design

Nearshore fishes were collected using (1) scuba and snorkeling, (2) beach seines, (3) plankton net, and (4) samples regurgitated from seabirds. Observations and *in situ* photography were also used to conduct species identification. Approval to conduct research on vertebrate animals was granted by the Russian Federation Ministry of Education and Science Approval Ref. No. 14-368 of 06.05.2013. Approval to conduct field studies was granted by the Russian Federation Ministry of Education and Science Approval Ref. No. 14-368 of 06.05.2013 permission to the Russian Arctic National Park.

### Diving methods

Three groups of divers conducted 68 dives at 19 localities at depths ranging from 0 to 34 m around the archipelago (Fig. 1). While diving, fish samples were collected using hand aquarium nets, nets and clove oil, or by hand. In addition, photographs were taken *in situ* to document underwater coloration and associated habitat.

### Beach seines

Beach seines were used at four locations: Tikhaya Bay at Hooker Island (40 hauls), Cape Tegetthoff at Hall Island (6 hauls), Nilsen Bay at Bell Island (4 hauls) and Phoka Bay at Northbrook Island (6 hauls). The seine was 10 m long by 2 m high with 10 mm stretch mesh.

### Plankton net

Larval and young-of-year fishes were collected by plankton net. Plankton sampling was conducted at 20 locations, primarily in the straits between islands at depths between 200 and 340 m, using vertical hauls from the bottom to surface. Plankton nets were of standard construction, with a mesh of 0.2 mm or 0.074 mm.

### Deepwater drop camera

To explore the deep sea environments, deep drop-camera surveys were conducted. National Geographic's Remote Imaging Team developed Deep Ocean Drop-cams, which are high definition cameras (Sony Handycam HDR-XR520V 12 megapixel) encased in a borosilicate glass sphere and rated to a 10,000 m depth. Viewing area per frame was between 2 and 6 m$^2$, depending on the steepness of the slope where the Drop-cam landed. Cameras were baited with 1 kg of frozen herring (Clupeidae) placed in a burlap bag and deployed for ca. four hours. The number of individuals of each taxa per drop-cam deployment was estimated from the maximum number of individuals observed per frame ($N_{max}$).

### Field processing of fish samples

Collected fishes were measured and weighed on board and preserved in 4% buffer formaldehyde solution. Samples were send to the Zoological Institute, St. Petersburg, Russia (entry number ZIN No 6-013) and examined by the senior author. Vertebra and fin ray number were counted on radiograms.

*Format of dates*: day, month, year.

*Sample number are as follows*: ZIN collection number (6-013) and fish identification number. For example, fish 22 is denoted as: ZIN 6-013/22.

## RESULTS

Our expedition identified sixteen species of fishes from seven families and three orders in the waters around FJL (Table 1). Of these, Scorpaeniformes was the most specious order, accounting for 81% of all species observed (Table 2). Species endemic to the Arctic accounted for 75% of the nearshore species observed (three-fourths of which had circumpolar distribution), followed by species with wider ranges, including the N. Atlantic and N. Pacific. Species with benthic or meso-benthic habitat preferences accounted for

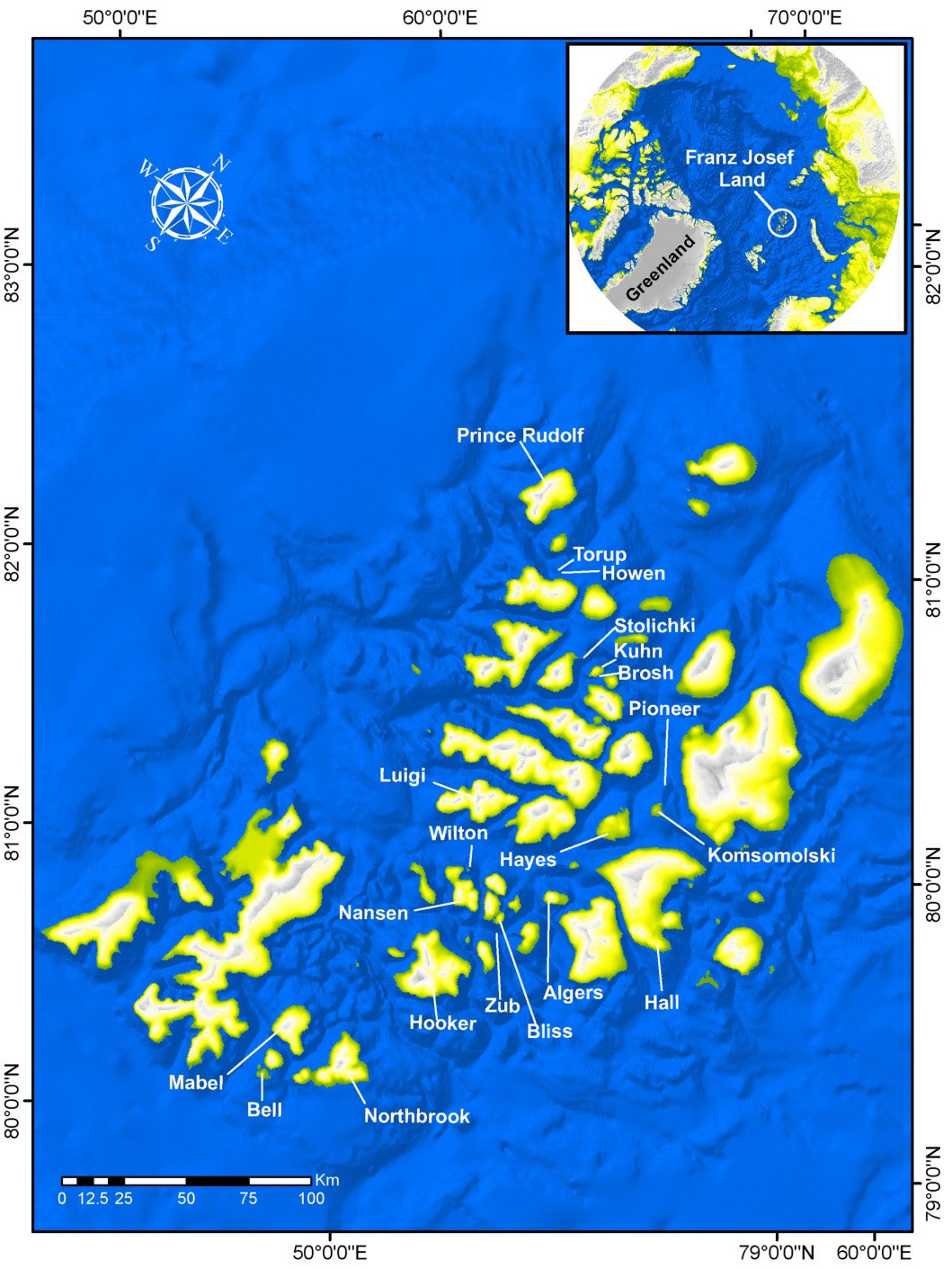

**Figure 1** Franz Joseph Land with names of islands where collections were conducted during the expedition.

**Table 1** Species of fishes observed during expedition to Franz Joseph Land in summer 2013.

| Common name | Scientific name | Trophic group | Habitat | Zoogeographic group[a] | Range[a] | Depth limits (preference), m[a] |
|---|---|---|---|---|---|---|
| **Somniosidae** | | | | | | |
| Greenland shark | *Somniosus microcephalus* | Pisc, Inv | Bentho-pelagic | high-boreal—arctic | Labrador, Baffin Bay to Kara Sea | 0–2200 (200–600) |
| **Gadidae** | | | | | | |
| Atlantic cod | *Gadus morhua* | Pisc, Inv | Bentho-pelagic | boreal–subarctic | W Atlantic—Barents Sea | 0–700 (100–200) |
| Polar cod | *Boreogadus saida* | Inv | Cryopelagic and bentho-pelagic | arctic | circumpolar | 0–1390 (0–500) |
| **Cottidae** | | | | | | |
| Twohorn sculpin | *Icelus bicornis* | Inv | Benthic | arctic | circumpolar | 0–930 (40–80) |
| Hamecon | *Artediellus scaber* | Inv | Benthic | arctic | circumpolar | 0–290 (0–50) |
| Bigeye sculpin | *Triglops nybelini* | Inv | Mesobenthic | arctic | circumpolar | 71–1354 (200–600) |
| **Cyclopteridae** | | | | | | |
| Atlantic spiny lumpsucker | *Eumicrotremus spinosus* | Inv | Bentho-pelagic | high-boreal–arctic | W Atlantic, Beaufort to Barents seas | 5–930 (60–200) |
| Derjugin's leatherfin lumpsucker | *Eumicrotremus derjugini* | Inv | Benthic | arctic | N Sea of Okhotsk-Arctic | 50–930 (50–275) |
| McAlpin's smooth lumpfish | *Cyclopteropsis mcalpini* | Inv | Benthic | arctic | Baffin Bay to Barents Sea | 50–170 |
| **Liparidae** | | | | | | |
| Parr's snailfish | *Liparis bathyarcticus* | Pisc, Inv | Benthic | arctic | circumpolar | 12–510 (30–350) |
| Kelp snailfish | *Liparis tunicatus* | Inv | Benthic | arctic | circumpolar | 0–415 (0–100) |
| Blackbelly snailfish | *Liparis* cf. *fabricii* | Inv | Benthic | arctic | circumpolar | 12–1460 (0–125) |
| **Agonidae** | | | | | | |
| Atlantic poacher | *Leptagonus decagonus* | Inv | Benthic | amphiboreal–arctic | W Pacific, W Atlantic, Arctic | 24–930 (120–475) |
| **Zoarcidae** | | | | | | |
| Arctic eelpout | *Lycodes reticulatus* | Inv | Meso-benthic | arctic | Circumpolar? | 20–930 (100–380) |
| Anderson's pout | *Gymnelus andersoni* | Inv | Benthic | arctic | Barents-Laptev seas | 28–300 |
| Esipov's pout | *Gymnelus esipovi* | Inv | Benthic | arctic | Barents-Laptev seas | 40–387 |

**Notes.**
[a] After *Chernova (2011)*; Pisc, piscivore; Inv, invertivore.

75% of the observed nearshore assemblage, followed by bentho-pelagic species (18.8%), and one cryopelagic and benthopelagic species, *Boreogadus saida*. FJL nearshore waters are at the upper limit of vertical distribution for fishes of all species. The vast majority of the fishes observed around FJL were invertebrate feeders, while three were facultative piscivores (e.g., Greenland shark, Atlantic cod, Parr's snailfish), with one—the cryopelagic polar cod—feeding primarily on zooplankton under the ice.

**Table 2  The number of fish orders, families, and species collected or observed in waters around Franz Josef Land during summer 2013.**

| Order | Family | Number of species |
| --- | --- | --- |
| Squaliformes | Somniosidae | 1 |
| Gadiformes | Gadidae | 2 |
| Scorpaeniformes | Cottidae | 3 |
| | Agonidae | 1 |
| | Cyclopteridae | 3 |
| | Liparidae | 3 |
| | Zoarcidae | 3 |

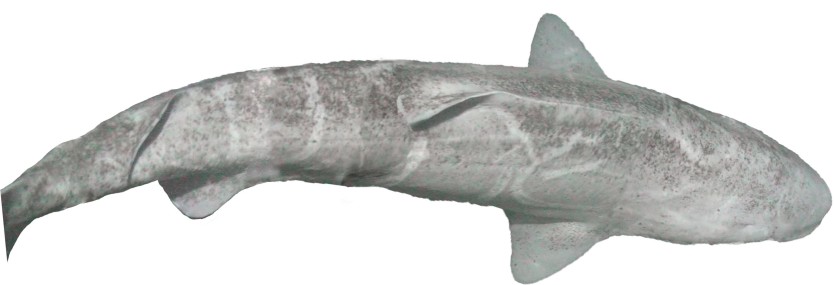

**Figure 2  Greenland shark, *Somniosus microcephalus*, off Hayes Island in 211 m.**

## Drop-cams

A total of 24 camera drops were conducted during the expedition, at depths from 32 to 292 m. Although numerous benthic organisms were observed (e.g., amphipods, soft corals, brittle stars, bryozoans) fishes were not common. Five fish taxa were observed during drop-cam surveys, with Polar cod the most common, occurring in 37.5% of the camera drops. Observations of Polar cod ranged from 125 to 392 m and averaged 0.75 individuals per drop, with $N_{max} = 6$ in any given frame. The cottid, *Iselus bicornis* was present on three drops (16.7%) with $N_{max} = 3$, and depths ranging from 58 to 132 m. Another cottid, *Artediellus scaber*, was observed on two drops with $N_{max} = 4$ at depths of 281–392 m. A single individual of an unidentified species, likely Capelin, *Mallotus villosus*, was observed twice; once at Torup I. (296 m) and once at Bell I. (121 m), but was not included in our species list due to taxonomic uncertainty. The most interesting sighting was one large (>2 m) Greenland shark (*Somniosus microcephalus*), which was observed off of Hayes Island on 13.08.2013 in 211 m.

## Species accounts

Family Somniosidae—Sleeper Sharks

*Somniosus microcephalus* (Bloch et Schneider, 1801)—Greenland shark (Fig. 2)

One 2 m specimen was recorded on drop camera off Hayes Island in 211 m (80°38.4N, 58°08.4E). This species was previously not known to occur around FJL (*Dolgov, 2011*;

 

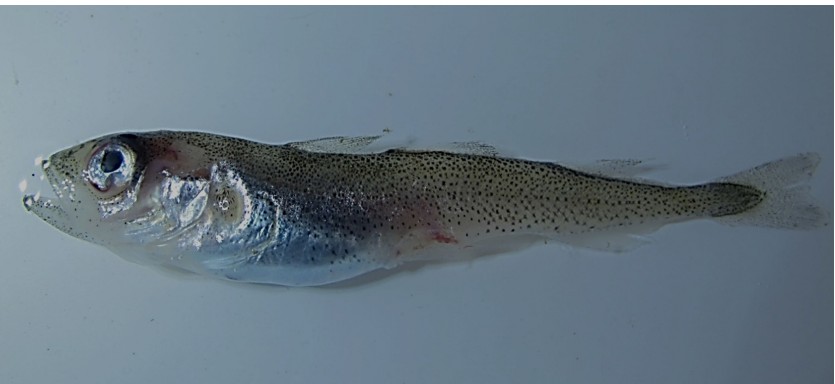

**Figure 3** **Polar cod, *Boreogadus saida*, TL 64 mm; Prince Rudolf Island.**

*Wienerroither et al., 2011*). This sighting therefore represents the first record of a Greenland shark from FJL and is the most north-eastern record for this species.

Family Gadidae—codfishes

*Boreogadus saida* (Lepechin, 1774)—Polar cod (Fig. 3)

Polar cod are common at FJL and were recorded at 9 islands at depths ranging from 6 to 21 m. Additional observations were made on drop cameras ranging in depth from 125 to 392 m. Polar cod occurred on 33% of the drop-cam deployments, averaging 0.75 individuals per frame, with a maximum of 6 in any given frame.

MC: ZIN 6-013/7, length TL 64 mm, Prince Rudolf I., Cape Fligeli, 81°50.92N, 59°15.77E, 19.08.2013, dp 6–21 m; habitat—stones, sand; coll. OV Savinkin, AN Chichaev.—ZIN 6-013/19, female TL 151 mm, 14.08.2013, Cape Podgorniy, Alger I., found on beach; coll. MV Gavrilo. VO: TL 160 mm, Hayes I., 80°37.82N, 58°03.29E, 12.08.2013, dp 11 m, A Friedlander. DC: observed on 8 of 24 camera drops at Algers (392 m), Hayes (171–211 m), Komsomolski (125 m), Kuhn (132 m), Luigi (357 m), Nansen (215 m), and Torup (297 m).

Polar cod were mentioned in FJL by numerous researchers dating back to the earliest expeditions (*Payer, 1878*; *Jackson, 1899*; *Nansen, Johansen & Nordahl, 1900*; *Collett & Nansen, 1899*; *Knipowitch, 1901*). Workers at the Tikhaya Bay Hydro-Meteorological station on Hooker Island (80.3°N, 52.8°E) in 1931–32 regularly observed polar cod, most often during ice-hummock formation (*Burmakin, 1957*). Explosives were used to collect these fishes at 10–15 m, resulting in >100 fish per blast event. In the summer of 1975 and 1979, the staff of the Hydro-Meteorological station at Hayes Island observed schools of polar cod "so large that they were scooped up by hand net" (*Borkin, 1983*). PINRO expeditions found polar cod were common south-east of the archipelago at depths of 170–460 m (*Borkin, 1993*). Other sources state that polar cod occur north towards the North Pole among the ice pack (*Andriashev, Mukhomediyarov & Pavshtiks, 1980*; *Mel'nikov & Chernova, 2013a*; *Mel'nikov & Chernova, 2013b*).

*Gadus morhua* Linnaeus, 1758—Atlantic cod

One Atlantic cod were collected. MC: ZIN 6-013/22, TL 200 mm, Alexandra Land, 80°42.58N, 47°31.55E, 28.08.2013; dp 15 m, habitat—rocks, coll. A Friedlander.

Previously, one cod larva (18 mm TL) was collected 28.08.1980 south-east of FJL (79°43′N, 46°28′E) by plankton net at 310–340 m; water temperature = −0.4 °C (*Borkin, 1993*). Adult cod were found south-west of FJL during the Russian-Norwegian Cooperation Program from 2004–2009 (*Wienerroither et al., 2011*).

Family Cottidae—Sculpins

*Icelus bicornis* (Reinhardt, 1840)—Twohorn sculpin (Fig. 4)

Individuals were recorded at 8 islands between 7 and 21 m.

MC: 8 specimens TL 30–95 mm from 5 stations; ZIN 6-013/2, juv. TL 32 mm, Hooker I., 80°19.14N, 52°48.66E, 02.08.2013, dp 20 m; coll. A Friedlander.—ZIN 6-013/5, female TL 48 mm, Kuhn I., 81°10.46N, 58°21.12E, 09.08.2013, coll. OV Savinkin.—ZIN 6-013/6, female TL 95 mm and 3 juv. TL 32–61 mm, Alexandra Land, 80°42.57N, 47°31.55E, 28.08.2013; ground—stones; dp 15 m.—ZIN 6-013/24, juv. TL 30 mm, Wilton I. (at the entrance to Esipov Bay on Nansen I.), 81°34.5N, 54°19.9E, 23.08.2013, st 31, dp 6–21 m, ground—boulders, stones, sand; coll. AN Chichaev.—ZIN 6-013/28, juv. TL 36 mm, Scott-Keltie I., 81°21.3N 52°26.0E, 27.08.2013, dp 17 m; ground—sand; coll. A Friedlander. VO: length to 130 mm, dp 7–15 m, A Friedlander. 1 sp, Northbrook I., Cape Flora, 79°57.43N, 50°02.67E, 05.08.2013.—1 sp, Brosh I., 81°06.27N, 58°21.01E, 08.08.2013.—3 sp, Kuhn I., 81°06.68N, 58°19.78E, 16.08.2013.—1 sp, Komsomol I., 80°38.82N, 58°55.05E, 17.08.2013.

The species is common at FJL. Fish were observed mainly in rocky areas, mixed with sand, boulders and shells, and water temperature <0 °C. Underwater observations show fish usually sitting on the bottom, perched atop widely spaced pectoral fins. A mottled disaggregated color provides camouflage over mixed bottoms.

*Artediellus scaber* Knipowitch, 1907—Hamecon (Fig. 5)

The species was found at 9 islands between 6 and 21 m.

MC: 9 specimens TL 31–75 mm. ZIN 6-013/14, TL 43 mm, Luigi I., 80°52.5N, 54°41.7E, 06.08.2013, dp 10 m.—ZIN 6-013/8, TL 62 mm, Prince Rudolf I., Cape Fligeli; 81°50.92N, 59°15.77E, 19.08.2013, dp 6–21 m; ground—small stones, sand; st 28, coll. OV Savinkin, AN Chichaev.—ZIN 6-013/16, TL 66 mm, Wilton I., 80°34.16N, 54°17.84E; 23.08.2013, dp 17 m; ground—rock; coll. A Friedlander.—ZIN 6-013/17, juv. SL 28 mm, Alger I., 80°22.79N, 53°46.18E, 14.08.2013, dp 12–22 m; ground—sand; st 18, coll. OV Savinkin.—ZIN 6-013/18, juv. TL 31 mm, Zub I. near Bliss I., 80°22.3 N, 54°39.6E, 24.08.2013, dp 11 m; ground—sand; st 32, coll. AN Chichaev,—ZIN 6-013/20, TL 75 mm, Kuhn I., 81°10.46N, 58°21.12E; 09.08.2013, coll. OV Savinkin.—ZIN 6-013/23, TL 73 mm, Wilton I., 81°34.5N, 54°19.9E; 23.08.2013; st 31, dp 6–21 m; ground—boulders, stones, sand, coll. AN Chichaev.—ZIN 6-013/27, juv. TL 28 mm, Scott-Keltie, 81°21.3N, 52°26.0E, 27.08.2013, dp 17 m; ground—sand; coll. A Friedlander.—ZIN 6-013/29, TL 69 mm, Prince Rudolf I., Teplitz Bay, 81°47N, 51°55E, 18.08.2013, dp 15–18 m, ground—boulders, sand; st 25, coll. SD Grebelniy. VO: 1 sp, Matilda I., 80°21.92N, 55°44.33E, 14.08.2013, 15 m, A Friedlander.—1 sp, Bliss I., 24.08.2013, OV Savinkin.

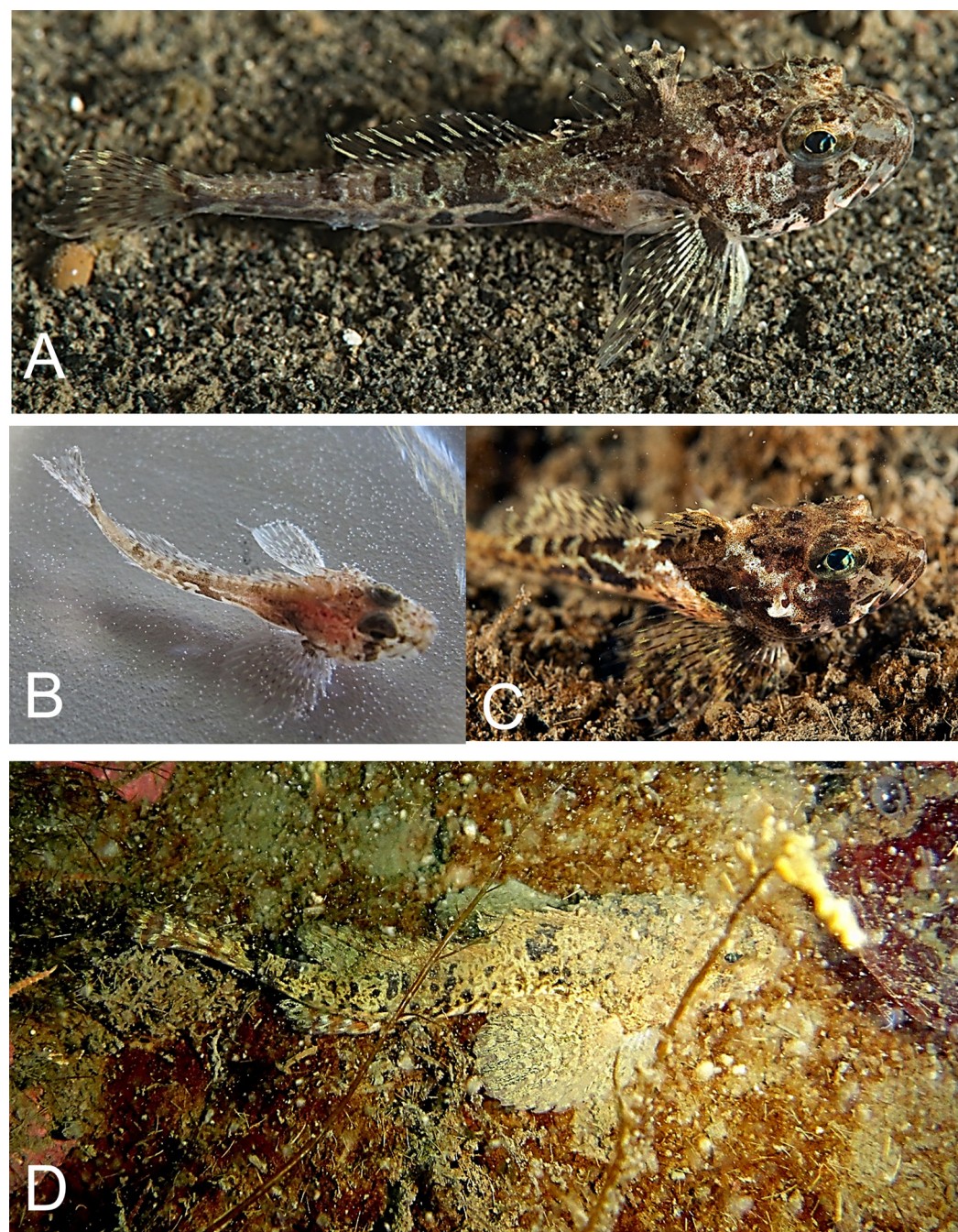

**Figure 4 Twohorn sculpin, *Icelus bicornis*.** (A) Young, Wilton Island; color is mottled, with dark brown spots and yellow strips with pale background. A large triangular spot is present below the first dorsal fin, with 4 small rounded spots are found below the second dorsal fin. Dark bands radiate around eye; fin rays locally yellow; a round brown spot is present on the base of pectoral fin. (B) Juvenile TL 34 mm, Hooker Island, ZIN 6-013/2. (C) Occipital thorns of hind pair are visible at nape, directed backward; narrow pale strip is present behind head, wide brown spot narrowing downward is present below the first dorsal fin. (D) Adult male, Kuhn Island, ZIN 6-013/5; dorsal and anal fins are enlarged, bright orange and white bands are intermittent at anal and caudal fins (sexual dimorphism).

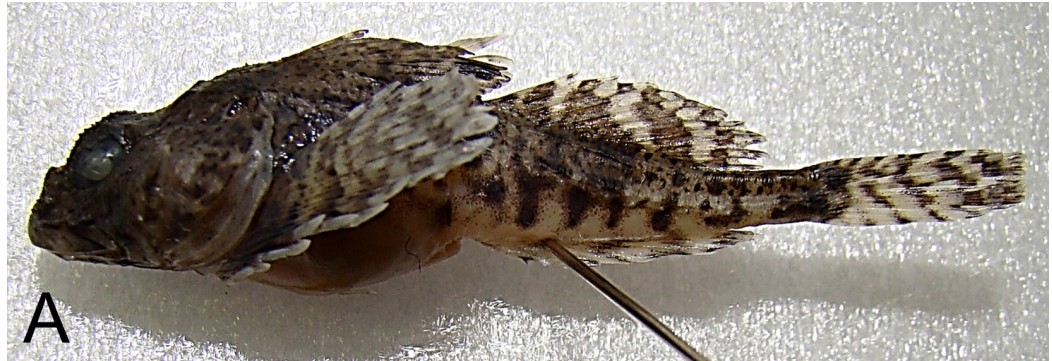

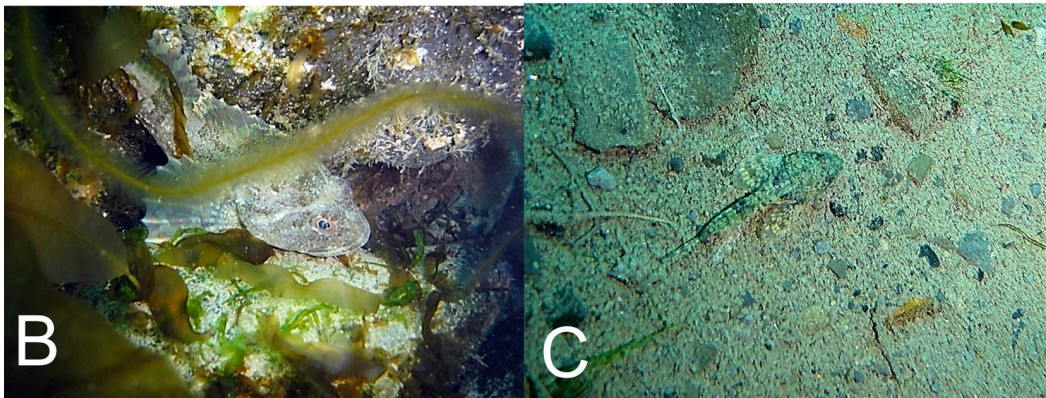

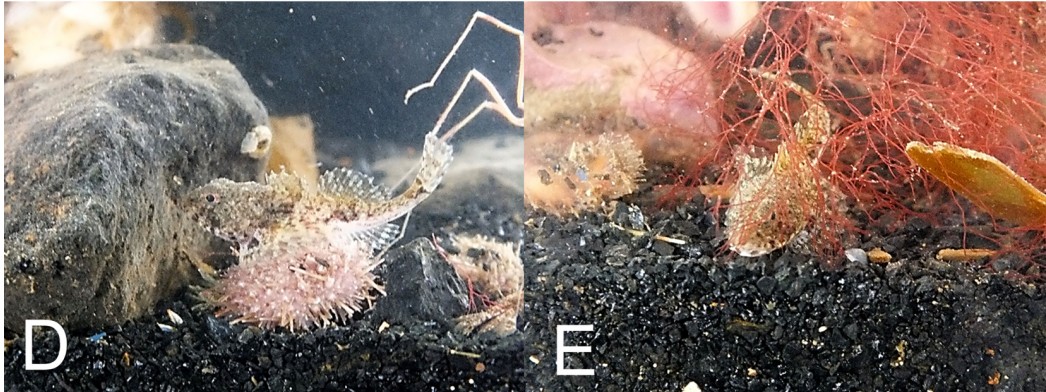

**Figure 5** **Hamecon,** *Artediellus scaber.* (A) Female TL 69 mm, ZIN 6-013/29, preserved specimen; nape behind eyes is depressed, occipital thorns not expressed. (B) Specimen is sitting near rock within area of macroalgae. (C) Mottled color camouflages fish on sandy-pebble bottom. (D) Hamecon feeding on peryphiton with sea urchin present in front; Bliss I. (E) Hamecon among red algae; Bliss I.

The Hamecon was a common inshore species and appeared to be highly site attached among rocks and boulders. A variegated color pattern was observed in mixed sand and rocky habitat. Hamecon were found in habitats ranging from the shallow red algae zone at Bliss Island to deeper (8–20 m) *Laminaria* spp. beds off Prince Rudolf Island (SD Grebelniy, pers. comm., 2014).

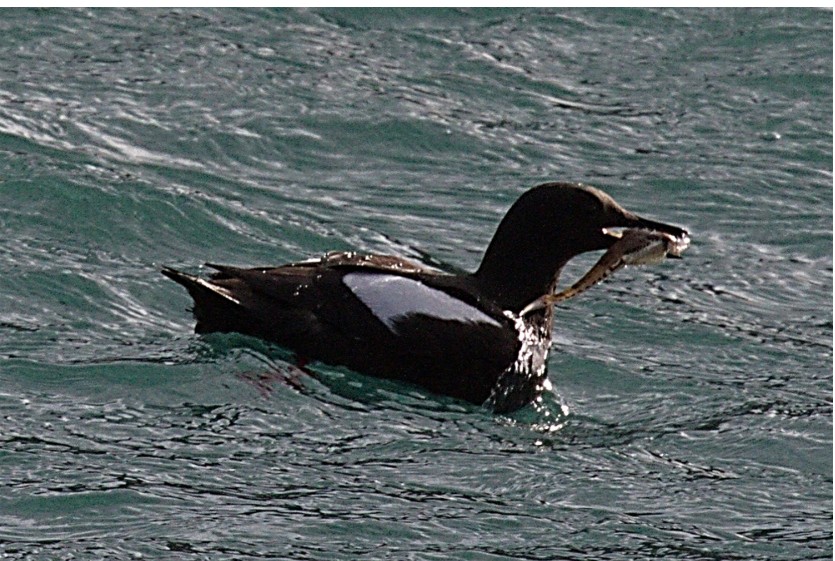

**Figure 6** Bigeye sculpin (*Triglops nybelini*) observed in the mouth of a Black guillemot (*Cepphus grylle*), Torup Island, 20.08.2013.

*Triglops nybelini*—Bigeye sculpin

One fish was identified by a photograph taken 20.08.2013 by MV Gavrilo at Torup I. (Fig. 6). The fish, ca. 15 cm TL, was in the mouth of a Black guillemot *Cepphus grylle* (family Alcidae). It was identified as a Bigeye sculpin, *Triglops nybelini*, based on its elongated body, wide pectoral fins, cottoid-like shape, and a line of black spots above the anal-fin base.

Bigeye sculpin are frequently caught in trawls around FJL, primarily over silty sand in 100–500 m (*Knipowitch, 1901*; *Andriashev, 1964b*). Hundreds of young-of-year (age 0+, 60–112 mm) were caught by pelagic trawl west of FJL (*Borkin, 1993*). The presence of two other *Triglops* species around FJL (*T. murrayi* and *T. pingelii*) (*Wienerroither et al., 2011*) need to be verified.

Family Cyclopteridae—Lumpfishes

*Eumicrotremus spinosus* (Fabricius, 1776)—Atlantic spiny lumpsucker (Fig. 7)

Specimens were recorded at 3 islands at depths from 8 to 15 m.

MC: 1 sp TL 31 mm, Matilda I. (opposite Alger I.), 80°21.92N, 55°44.33E, 14.08.2013, dp 8–15 m; A Friedlander. VO: adult, Wilton I., 23.08.2013, photo: OV Savinkin;—adult, Bliss I., 24.08.2013, photo in aquarium: AP Kamenev;—juv., 24.08.2013, same location, photo: A Friedlander.

Adult fish were observed adhering to rocks using ventral disks, juveniles were found sitting on thalli of *Laminaria* spp. kelp. Previously, this species was recorded in FJL at 79°55N, 49°48E, 27.07.1901, in 34 m over shell habitat (*Knipowitch, 1901*), and also found west of Alexandra Land (*Wienerroither et al., 2011*).

*Eumicrotremus derjugini* Popov, 1926—Derjugin's leatherfin lumpsucker (Fig. 8)

Specimens were collected at Prince Rudlof and Zub islands at 6–21 m.

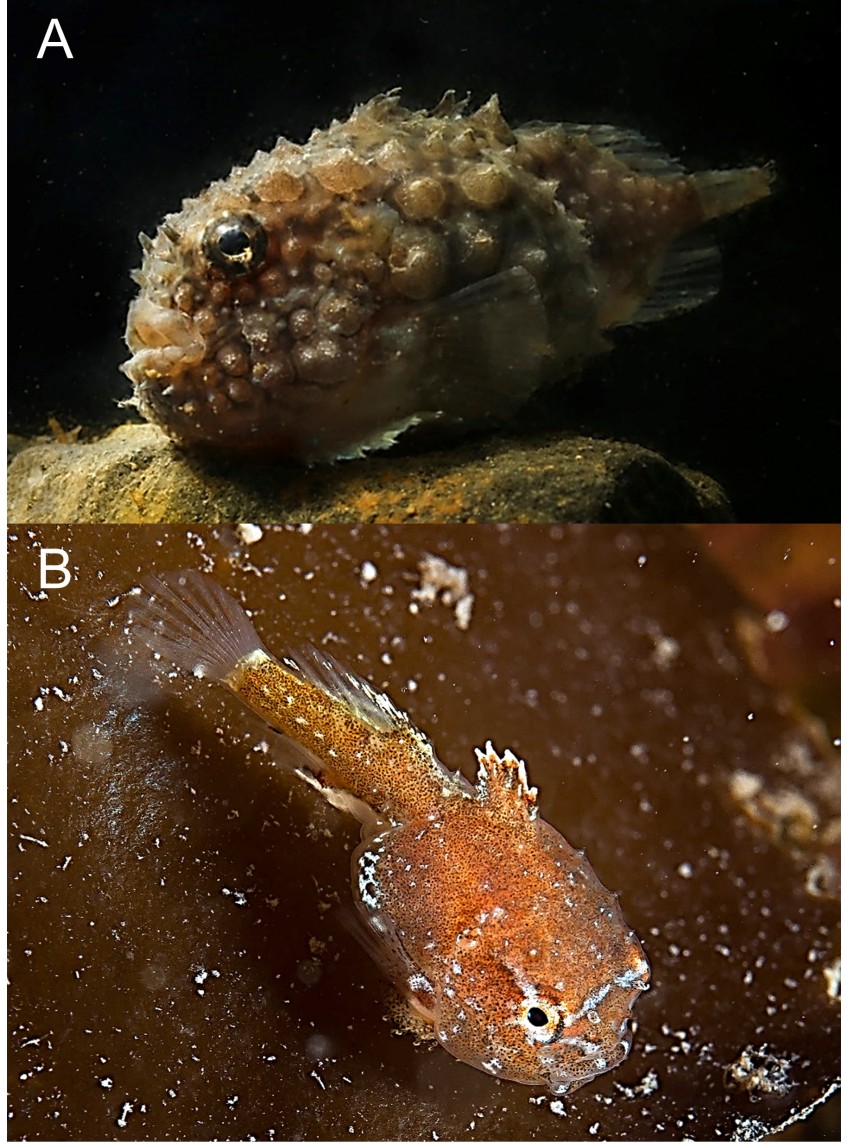

**Figure 7** **Atlantic spiny lumpsucker, *Eumicrotremus spinosus*.** (A) Bliss Island, body is entirely covered by spiny plates; two pair of tubular nostrils are visible on snout in front of eye; barbels on chin present. (B) Juvenile sitting on *Laminaria* sp.; caudal portion is nearly equal to trunk length.

MC: ZIN 6-013/10, juv. TL 14 mm, Prince Rudolf I., Cape Fligeli, 81°50.92N, 59°15.77E, 19.08.2013, dp 6–21 m; ground—small stones, sand; st 28, coll. OV Savinkin, SD Grebelniy. VO. Young, 24.08.2013, Zub I., photo: OV Savinkin.

  *Eumicrotremus derjugini* differs from *E. spinosus* by having the first dorsal fin hidden in thick fleshy leather; spiny plates on the body are smaller; enlarged plates in front of the anal fin origin and along the first dorsal fin are absent; transversal skin folds between ventral sucking disk and anus are absent; barbels on chin are absent. In young fish, spiny plates are less developed than in adults. Previously a single juvenile was recorded from a trawl (79°46′N, 63°08′E) at 240 m (*Borkin, 1983*; *Borkin, 1993*).

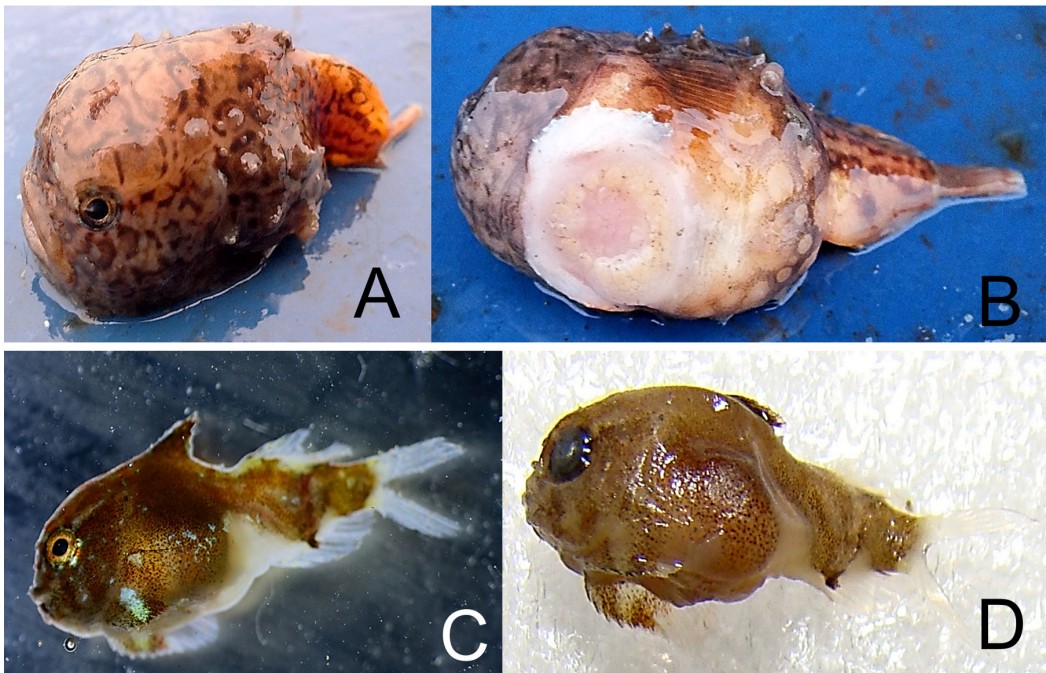

**Figure 8 Derjugin's leatherfin lumpsucker, *Eumicrotremus derjugini.*** (A) Young, Zub Island; forehead prominent, snout abruptly vertical; spiny plates weakly developed and small, appearing in three longitudinal short rows, 3–4 plates each; color is light brown with dark twisting lines. (B) Same specimen from below; sucking disk consists of transformed ventral fins. (C) Juvenile TL 14 mm, Prince Rudolf Island, ZIN 6-013/10; the first dorsal fin already covered by thickened skin. (D) Same specimen after preservation.

*Cyclopteropsis mcalpini*—McAlpin's smooth lumpfish (Fig. 9)

An adult *Cyclopteropsis mcalpini* ca. 40 mm TL was observed sitting on the empty shell of the Gastropod *Neptunea* sp., with its egg mass inside the shell. This shell was collected between Torup and Howen islands at 81°31N, 58°31.7E, 20.08.2013, st 29, dp 18–31 m, habitat—rock, stones, sand, shells; coll. OV Savinkin.

Forehead is wide and flattened, mouth is up turned; a row of 4 small spiny plates present on body sides; color is pale with dark brown irregular net-like spots. The eggs, ca. 6 mm in diameter, had well developed larvae that were nearly ready to hatch. *Andriashev (1964a)* noted parental care in *Cyclopteropsis* when a male was found on an empty gastropod shell protecting juveniles.

Family Liparidae—Snailfishes

*Liparis bathyarcticus* Parr, 1931—Parr's snailfish (Fig. 10)

MC: 5 specimens from 2 stations; ZIN 6-013/3, TL 41 mm, Hayes I., 80°37.63N, 58°03.88E, 13.08.2013, dp 8 m; coll. A. Friedlander.—ZIN 6-013/33, 5 juv. TL 15–16 mm, Pioneer I., 80°38.82N, 58°55.05E, 17.08.2013, dp 15 m; st 22; coll. A Friedlander. VO: 2 specimens at Hayes I., Cape Druzhniy, 80°37.64N, 58°04.104E, 13.08.2013, dp 6–1.5 m.

Parr's snailfish (ZIN 6-013/3) were found at 8 m, in rocky habitat with gravel and clay. A pair was observed sitting in a hole between rocks among small brown and green algae,

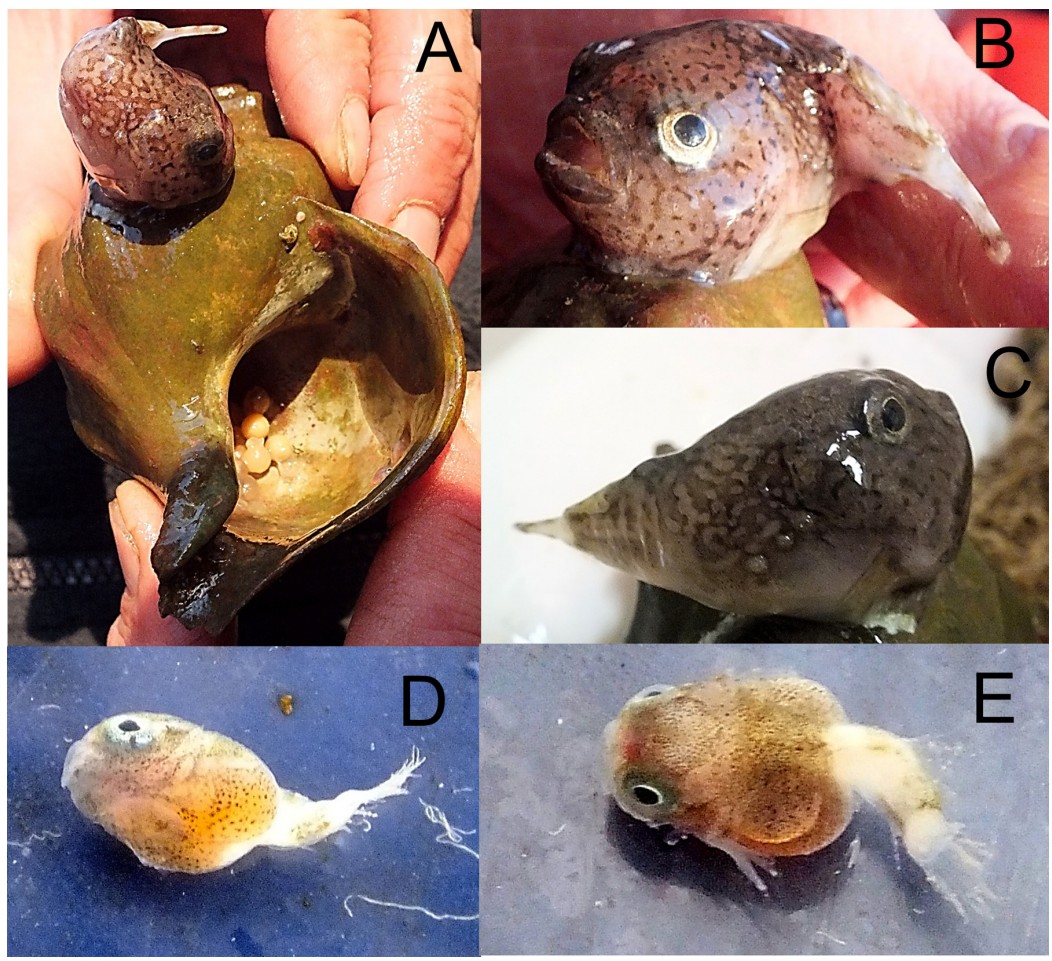

**Figure 9** **McAlpin's smooth lumpfish,** *Cyclopteropsis mcalpini,* **collected between Torup and Howen islands.** (A) Adult 30 mm in length; sitting on *Neptunea* sp. shell; egg clutch is visible inside the shell. (B, C) Close-up of same specimen. (D, E) Embryo, extracted from egg, which was attached to *Neptunea* sp. shell; yolk sac is large, sucking disk present.

at 1.5–6 m. Larval *L. bathyarcticus* were observed 17.08.2013 in large numbers at 10–15 m at Pioneer Island. Densities were as high as 10 s m$^{-2}$. Similar densities were observed on nearby Kuhn Island between 6 and 10 m on 16.08.2013 (A Friedlander). Mean length of larvae was 13 mm ($n = 12$) and were without pigment.

Early stage juveniles (TL 15–16 mm; egg sack still present; sucking disk entirely developed) were present in our samples. The gill slit is as in adults, large and reaching down to mid-base of pectoral fin upper lobe (in other arctic snailfishes it only extends from the 1st to 6th pectoral fin ray). The name *L. bathyarcticus* was revalidated (*Chernova, 2008*). Adults feed partly on fishes.

*Liparis tunucatus* Reinhardt, 1837—Kelp snailfish (Fig. 11)

We recorded twelve specimens (TL 46–166 mm) at four islands between 6 and 30 m.

MC: ZIN 6-013/9, female TL 137 mm, Prince Rudolf I., Cape Fligeli, 81°50.92N, 59°15.77E, 19.08.2013, dp 6–21 m; ground—small stones, sand; st 28, coll. OV Savinkin,

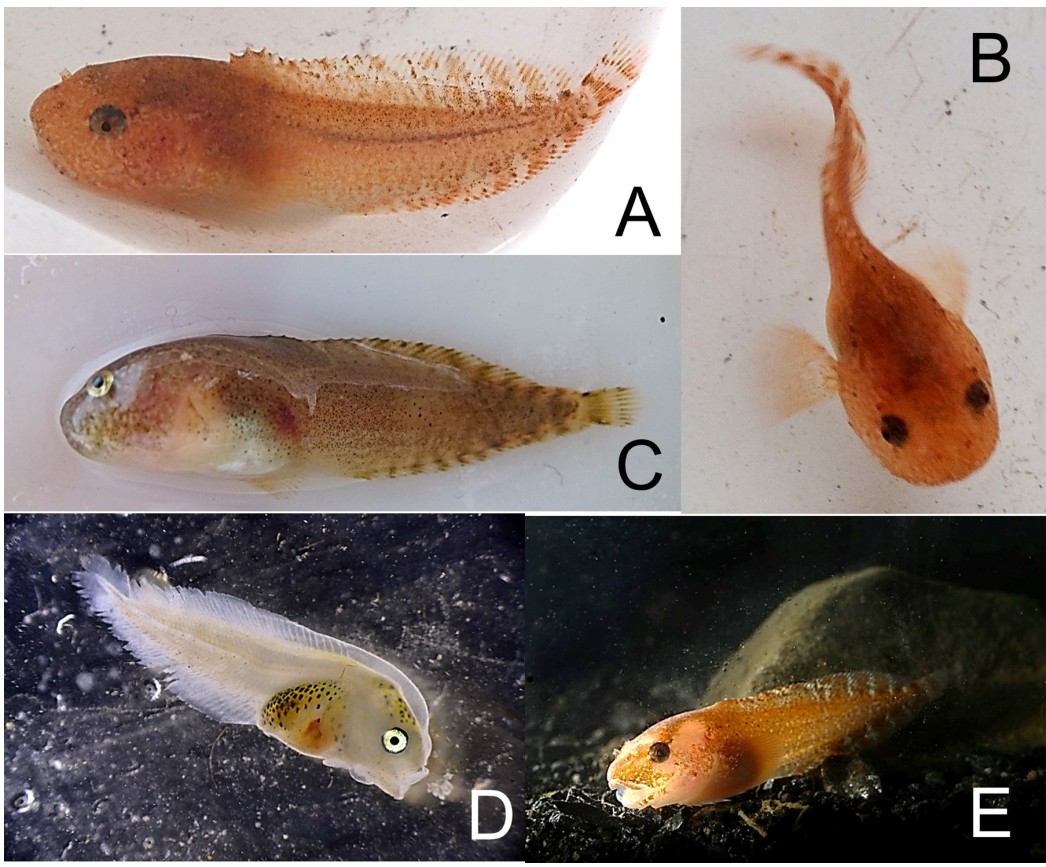

**Figure 10** **Parr's snailfish, *Liparis bathyarcticus*.** (A, B) Young specimens alive, TL 41 mm, Hayes Island, ZIN 6-013/3; color orange-brownish. (C) Same specimen; color brown after death; (D) Juvenile TL15 mm captured in plankton net; Pioneer Island, ZIN 6-013/33. (E) *In situ* specimen TL 41 mm.

AN Chichaev.—ZIN 6-013/12, juv. TL 46 mm, Prince Rudolf I., 81°51.09N, 59°14.76E, 19.08.2013, dp 15 m, coll. A Friedlander.—ZIN 6-013/11, male TL 145 mm, and 3 sp, length 105–109 mm; Stolichki I., eastward of Milovzorov rocks, 81°11.91N, 58°11.46E, 08.08.2013; dp 6 m; coll. A Friedlander. VO: 6 sp, length to 166 mm, A. Friedlander. 3 sp, Prince Rudolf I., 81°51.09N, 59°14.76E, 19.08.2013, dp 15 m.—2 sp, Howen I., 81°30.95N, 58°21.41E, 20.08.2013, dp 15 m.—1 sp., Nansen I., 80°34.16N, 54°17.84E, 23.08.2013, dp 30 m.

Kelp snailfish are common at FJL. Divers in 1981–1982 collected 47 specimens from 24 localities at 1–32 m (*Chernova, 1989*; *Chernova, 1991*; *Chernova, 1993*). Fish were dark red and usually attached to lower surface of kelp thalli, or under rock using their sucking disks. Spawning is known to occur in March, and previous underwater observations found blackish-green egg clutches on kelp thalli at 6 to 25 m during this time.

*Liparis* cf. *fabricii* Krøyer, 1847—Blackbelly snailfish (Fig. 12)

Specimens were found by divers at 6 islands between 10 and 25 m; fry SL 31–83 mm were collected by plankton net between 142 and 400 m.

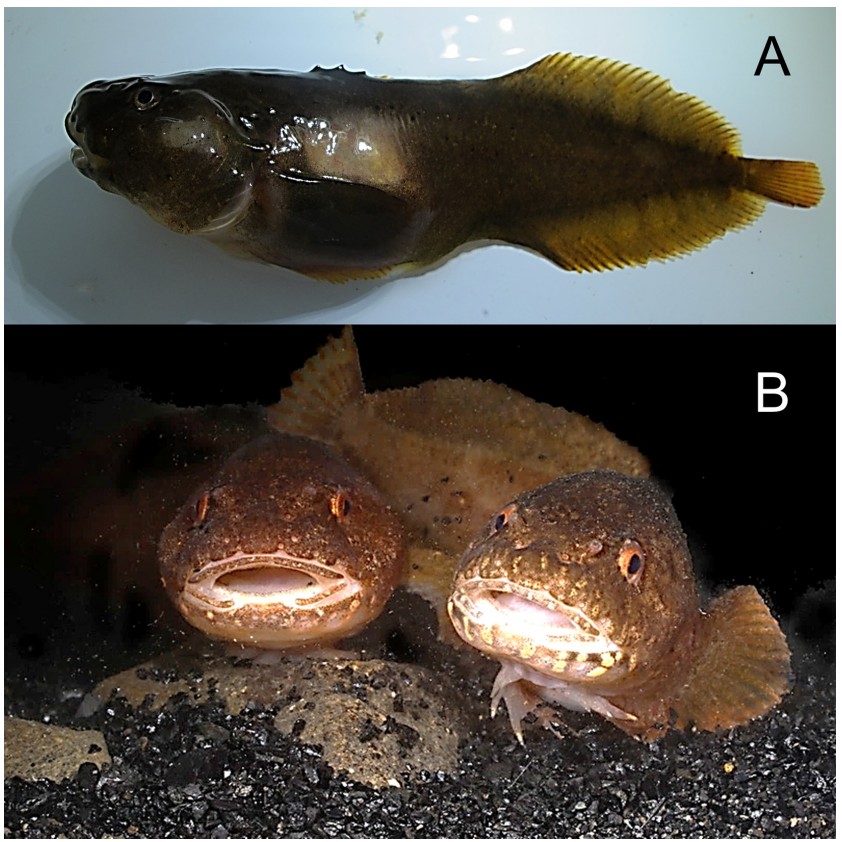

**Figure 11** **Kelp snailfish,** ***Liparis tunicatus.*** (A) Female TL 137 mm, Prince Rudolf Island, ZIN 6-013/9. (B) A pair *in situ*, Stolichki Island, ZIN 6-013/11.

MC: 5 sp TL to 165 mm; ZIN 6-013/13, male TL 165 mm, Luigi I., Burke Bay, 81°51.72N, 54°47.30E, 6.08.2013, dp 20–25 m; ground—silty sand, stones, silt; fish sitting on *Laminaria*; coll. AN Chichaev.—ZIN 6-013/25, TL 106 mm, Bell I., 80°02.29N, 49°11.75E, 21.08.2013, dp 17–22 m; ground—sand; coll. OV Savinkin.—ZIN 6-013/26, 2 sp TL 72 mm, Prince Rudolf I., 19.08.2013; plankton net between 200 and 340 m; coll. DM Martinova.—ZIN 6-013/32, 1 sp SL 31 mm, straight near McClintock I., 24.08.2013, 80°04.7N, 55°24.7E.; plankton net between 142 and 400 m; coll. A Friedlander. VO (OV Savinkin): young specimen, Mabel I., 25.08.2013;—juv, Hooker I. at Rubini Rock, 1–2.08.2013.

The blackbelly snailfish *Liparis fabricii* is a species complex which differs from other Arctic snailfishes in that it has a black peritoneum (i.e., wall of body cavity) (*Chernova, 2008*).

The form from FJL collected by our expedition had a rounded head (head width is equal to head depth) and a tapered snout with a prominent point. The posterior nostril is half the size of the anterior nostril, without flap-like projections. The mouth is horizontal with small anterior teeth, jaws are trilobate, posterior teeth have small lateral shoulders. Snout folds are undeveloped, opercular flaps are rounded. Gill slits reach to 5–8th pectoral rays. Some anterior dorsal fin rays are shorter than posterior rays. The upper pectoral fin lobes

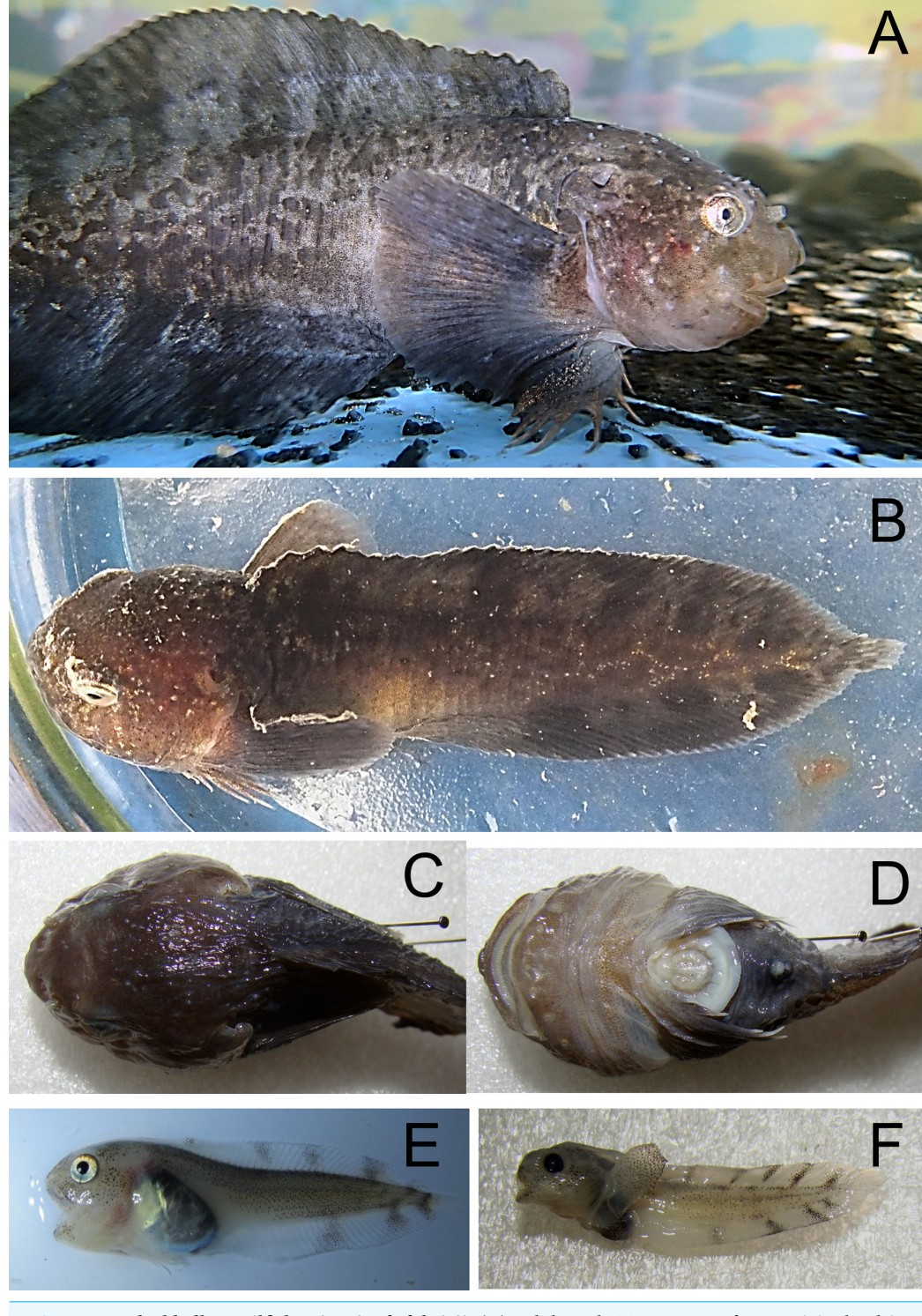

**Figure 12 Blackbelly snailfish, *Liparis* cf. *fabricii*.** (A) Adult male TL 165 mm from Luigi Island in aquarium, ZIN 6-013/13; note elongated pectoral fin lower-lobe rays with tactile receptors. (B) Specimen from Mable Island in an aquarium. (C, D) Length TL 106 mm, view from above and below; Bell Island. (E) Young specimen TL 72 mm; collected in plankton net; dark spots are distinct on transparent fins; black peritoneum hidden by silvery pigmentation. (F) Young TL 33 mm after preservation, ZIN 6-013/ 32.

reach slightly behind the anal fin origin. The disk is well developed and the skin of adult males is covered by cone-like prickles. Vertebrae 50–53 (11 + 39 − 42); D 44–49, A 37–41; P 36 (28 + 8). Caudal fin includes 9–10 principal rays, 2 upper and 2 lower secondary rays. Color is blackish with small black spots; between 3 and 5 wide oblique bands present at dorsal and anal fins. Peritoneum black, after preservation without silvery pigmentation.

Pelagic young have a dark band along the dorsal fin base; dorsal and anal fins are semi-transparent, with 5 and 3 transversal blackish spots, respectively; the peritoneum is black and distinctly visible through body wall, masked from the outside by silvery guanine pigmentation that quickly disappears after preservation in formaldehyde.

Underwater observation shows adults close to the bottom, among thalli of kelp, at a depth 10–25 m; ground—silty sand with stones, overgrown by algae (mainly *Sacharina latissima* and *Alaria esculenta*).

Blackbelly snailfish are common at FJL. Divers from the MMBI expedition collected 20 specimens at 13 locations (*Chernova, 1993*). Based on PINRO surveys, *L. fabricii* likely school, with >500 specimens caught in a one hour trawl; aggregations of young in pelagic trawls (0+, 100 s of specimens) were recorded westward of FJL, at 81°04N, 43°18E (*Borkin, 1993*).

Family Agonidae—Sea Poachers

*Leptagonus decagonus* (Bloch et Schneider, 1801)—Atlantic poacher

One Atlantic poacher, TL 180 mm, was identified from Brosh Island (81°06.28N, 58°21.19E), 08.09.2013, at 10 m (identification: A Friedlander). *Leptagonus decagonus* (16–21 cm) were collected at FJL by PINRO surveys at 245–400 m (*Borkin, 1993*), and also south of Alexandra Land (*Wienerroither et al., 2011*).

Family Zoarcidae—Eelpouts

*Lycodes reticulatus* Reinhardt, 1835—Arctic eelpout (Fig. 13)

Three specimens between 180 and 48 mm TL were recorded at depths from 6 to 15 m sheltering in rocky areas among kelp (identification: A. Friedlander): Brosh I., 81°06.26N, 58°21.15E, 16.08.2013, dp 15 m.—Hayes I., 80°37.82N, 58°03.29E, 12.08.2013, dp 6 m.—Howen I., 81°30.95N, 58°21.41E, 20.08 2013, dp 15 m. Previously specimens were collected from Alexandra Land at 124 m (*Andriashev, 1964b*), and south of the archipelago between 180 and 410 m (*Borkin, 1993*; *Wienerroither et al., 2011*).

*Gymnelus andersoni* Chernova, 1998—Anderson's pout (Fig. 14)

MC: ZIN 6-013/1, male TL 120 mm, Hooker I., Tikhaya Bay, 80°19,39N, 52°50,87E, 4.9.2013.

One specimen was found on the deck of the ship partially digested and likely deposited by a seabird (caudal portion missing). The specimen has infraorbital pores absent; the origin of the dorsal fin was near vertical of the anal-fin origin; pre-dorsal distance (from tip of snout to dorsal fin origin) is 205% of head length (lc). Radiogram counts: abdominal vertebrae 21. The first ray of D-fin located between vertebrae 14 and 15; 11 interneuralia bear no corresponding dorsal fin ray, the first is between vertebrae 4 and 5. Head depressed. Pectoral fin length 57.5% lc. Pectoral-fin rays 10.

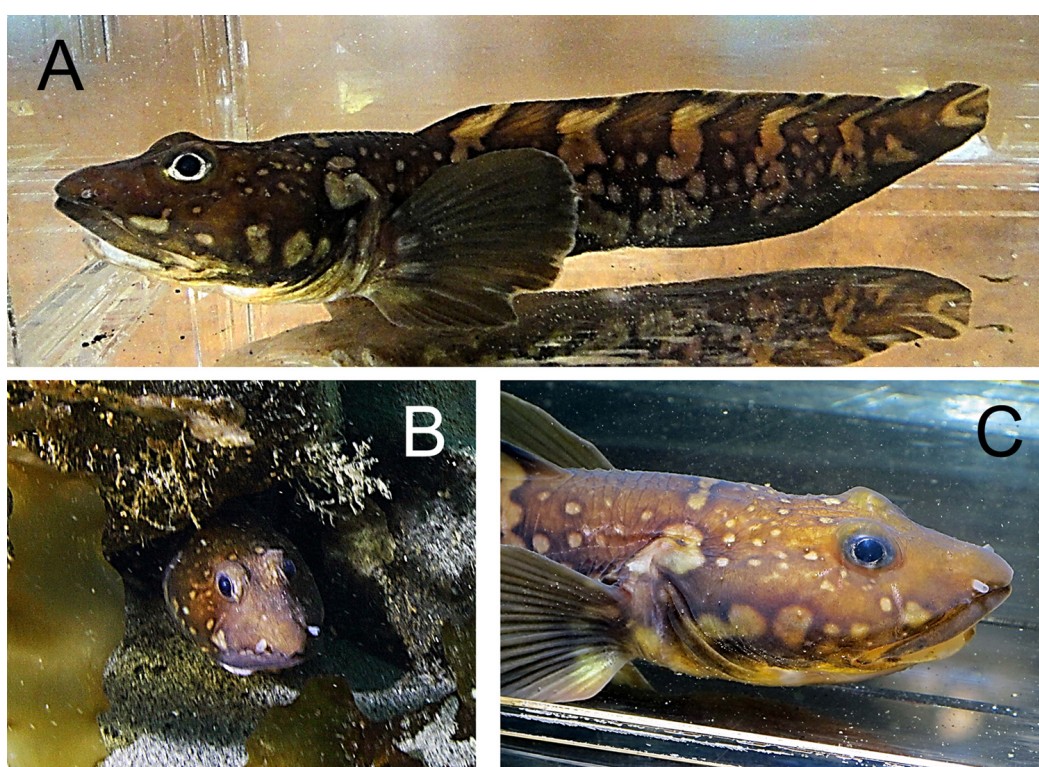

**Figure 13 Arctic eelpout, *Lycodes reticulatus*, Hayes Island** (A) Specimen in aquarium, color is dark, brown-black, bright light spots and transversal bands are present. (B) Arctic eelpout *in situ*, hidden in a shelter under rocks. (C) Same aquarium specimen as above.

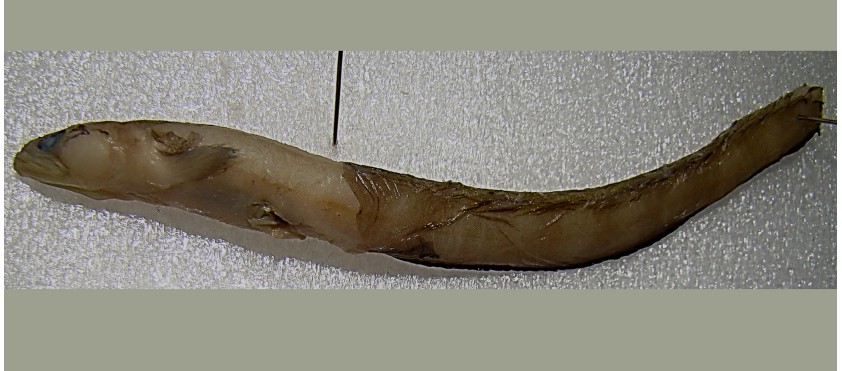

**Figure 14 Anderson's pout, *Gymnelus andersoni*, male TL 120 mm, ZIN 6-013/1.** Specimen found on ship's deck, partially digested and likely deposited by a seabird. Dorsal fin origin near dissecting needle. Buccal muscle enlarged and prominent (sexual dimorphism). Indistinct vertical light brown and pale wide bands alternating on body.

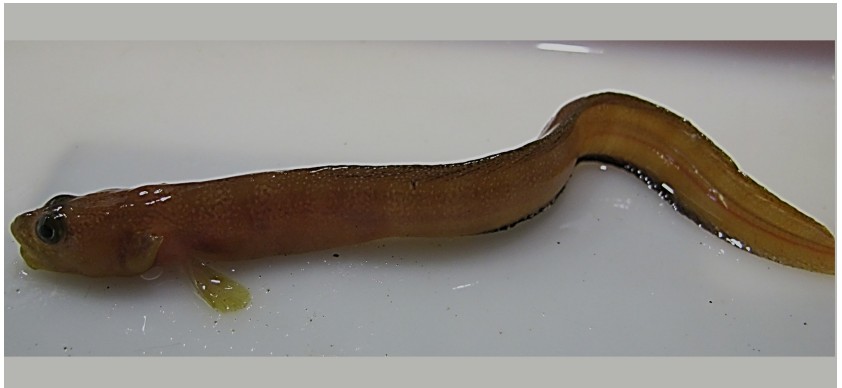

**Figure 15** Esipov's pout, *Gymnelus esipovi*, male TL 141 mm, Wilton Island.

Anderson's pout differs from *G. viridis* by the posterior position of dorsal fin origin, with the pre-dorsal distance 2x longer than the head length. *Gymnelus andersoni* differs from *G. retrodorsalis* (with similar position of dorsal fin origin) by having reduced sensory pores in the infraorbital canal (pores below eye are closed or entirely absent) (*Chernova, 1998a*; *Chernova, 1998b*). Paratypes of *G. andersoni* derive south-east of FJL (77°31N, 64°34E) at a depth of 280 m (ZIN 14160).

*Gymnelus esipovi* Chernova, 1999—Esipov's pout (Fig. 15)

MC: ZIN 6-013/4, male TL 141 mm, Wilton I., 80°34.16N, 54°17.84E, 23.08.2013, dp 15 m, habitat—rock; collected in kelp by A Friedlander.

*Gymnelus esipovi* differs from *G. andersoni* in its anterior dorsal fin position (pre-dorsal distance <1.3 larger than head); infraorbital pores are well developed. This species was previously known only from Spitsbergen, the northern Barents Sea near Novaya Zemlya (75°53-76°30N, 51°55-57E), and the northern Kara Sea (*Chernova, 1999a*). This is the first record of *Gymnelus esipovi* from FJL.

## DISCUSSION

### FJL fish fauna diversity

Our expedition identified 16 species of fishes from 7 families mainly in nearshore FJL (<34 m). We added two species previously unknown for FJL; the Greenland shark (Hayes Island, 211 m) and Esipov's pout (Wilton Island, 15 m). For many other species, our expedition increased the numbers of localities within the archipelago where these species are known, as well as identifying new upper depth limits for many species. The only nearshore fish species previously recorded from the archipelago but not found during our expedition was the tape-body pout *Gymnelus taeniatus*, which is likely locally endemic to Franz Josef Land (*Chernova, 1999a*).

Most of the trophic diversity is comprised of invertebrate feeders while three species (Greenland shark, Atlantic cod, and Parr's snailfish) are facultative piscivorous. However the vast majority of the fish biomass for the region is likely polar cod which can feed on

zooplankton under the ice. The low observed abundance of this species may be explained by the lack of sea ice near FJL during our expedition.

Species endemic to the Arctic accounted for three quarters of the nearshore species, the majority of which are common at high latitudes and circumpolar in the Arctic (*Chernova, 2011*). The distribution of some species (e.g., Arctic eelpout, Anderson's pout, Esipov's pout, and McAlpin's smooth lumpfish) is currently not fully known. The presence of larvae and young-of-year of many of the nearshore fishes suggests as least some local spawning and self-recruitment. The nearshore waters around FJL are important as nursery habitat for a number of fishes found in deeper water.

Compared with much of the Barents Sea, the fish fauna of FJL is depauperate. The waters around FJL are at or below freezing year round and the littoral zone is covered by ice during much of the year. As a result, nearshore fishes are restricted to a zone between 6 and 25 m, in areas dominated mainly by macrophytes.

Several species may only spend a portion of their lives in FJL. The Greenland shark *S. microcephalus* is nomadic. FJL is not recognized as a nursery or feeding area for young Atlantic cod and capelin, but favorable currents and oceanographic conditions may allow these species to survive until they migrate to spawning areas towards the south and west. The Arctic cod, *Arctogadus glacialis*, is a cryopelagic (sympagic) species that occurs south of FJL, as well as northward to the North Pole (*Andriashev & Chernova, 1994*; *Chernova, 2011*). The lack of pack ice around FJL during our expedition may account for the absence of this species from our list. Some mesopelagic fishes are also non-residents. Glacier Lanternfish *Benthosema glaciale* (*Borkin, 1986*) and White Barracudina *Arctozenus risso* occur south and west of the archipelago, but the extreme environment conditions of FJL likely prevent these species from reproducing in these waters. Many of the other species that are known from FJL occur in >34 m and therefore not encountered on our surveys.

Trawl catches around FJL in depths 100–600 m have recorded 43 fish species from 15 families. Most are primarily demersal, non-migrating species with the exception of the commercially important black halibut, which spawns on the western continental slope of the Barents Sea. Nursery and feeding areas for black halibut occur west of FJL in the Franz Victoria Trough, as well as in the Voronin and St. Anna troughs to the east. With the exception of black halibut, there are currently no fish species in commercially-exploitable abundance around FJL.

## Abundance of FJL fishes

Notes published from the Jackson–Harmsworth Expedition of 1894–97 state: "Though angling with line and hook was tried, it proved unsuccessful; and in order to obtain specimens of fish, it became necessary to stand on the shore and wait for the birds that came flying in from the distant open water with fish in their mouths. These birds were promptly shot, and came tumbling down with the fish still in their grip. This is an instance, I fear, of highway robbery with violence to the person, but science condones much". Later in the text they state: "Fishing with line met with no success, but many specimens were taken from birds, and are preserved and brought back" (*Brice & Fisher, 1896*).

Our results confirm that fishes in FJL are not abundant. The density of fishes observed during dives was very low. On average, only a few individuals were observed on any dive ($N = 68$). These low densities likely reflect very low standing stock of benthic fishes, but also related to the fact that many of the individuals were extremely cryptic, occurring on the underside of kelp fronds or hiding within rocks and/or kelp holdfasts. Underwater visual surveys of fishes off West Greenland yielded densities of 0.03 fishes m$^{-2}$ (*Gremillet et al., 2004*) and are typical for other Arctic rocky shores (*Hoff, 2000*; *Born & Böcher, 2001*). Results of our beach seine efforts confirm fish scarcity. In 56 hauls at 4 locations no fishes were collected and no fishes were observed in this habitat. There was continuous daylight during the entire expedition so it is unclear if the abundance patterns we observed are similar during other times of the year. However due to sea ice conditions during the winter months, abundances are likely highest during the summer period when our sampling was conducted.

The blood serum of most fishes freezes at $< -0.07\,°C$ (*Holmes & Donaldson, 1969*) and therefore the shallow water or extremely cold, ice-laden deep waters surrounding FJL are inhospitable for most species. Fish that live in the polar oceans survive at low temperatures by virtue of 'antifreeze' plasma proteins in the blood that bind to ice crystals and prevent these crystals from growing (*Fletcher, Hew & Davies, 2001*; *Marshall, Fletcher & Davies, 2004*). The blood of the cryopelagic fishes, such as Notothenioids *Pagothenia borchgrevinki* and *Dissostichus mawsoni* (Perciformes) in Antarctic and cod species *B. saida* and *A. glacialis* in the Arctic, contains glycoproteins that serve as antifreeze agents.

No commercial fishing occurs in FJL due to extensive ice cover for most of the year and the absence of commercially abundant fishes. The Atlantic cod, haddock, capelin, red-fishes, wolf fishes, and flat fishes either are absent from FJL or occur at densities too low for commercial exploitation. Polar cod are common in FJL waters, with enormous shoals sometimes observed, but the low value of this species has precluded commercial exploitation. Polar cod are important ecologically in high-Arctic areas. They are an essential link in the cryopelagic food web by grazing on under-ice zooplankton and in turn are a major food component of many seabirds and marine mammals (*Bradstreet et al., 1986*; *Finley, Bradstreet & Miller, 1990*; *Welch, Crawford & Hop, 1993*). Many questions on the biology and ecology of arctic fish still remain unanswered.

## Trends assuming climate change scenarios

The climate projections for the eastern Arctic show a warming that will cause a shift from sea-ice algae-benthos-dominated to zooplankton-dominated communities (*Bates & Mathis, 2009*). Such a fundamental shift may have negative consequences on large marine carnivores (e.g., seabirds and marine mammals), but have a positive influence on the abundance of smaller carnivores (e.g., fishes), because the average body size of prey will decrease substantially (*Karnovsky et al., 2003*).

Until recently, the north-eastern Barents Sea has had permanent ice-cover, but during the last decade the entire shelf sea has been ice-free during the summer months (*Johannesen et al., 2012a*). A retreat and thinning of the ice cover in the Barents Sea likely

will result in the northern portion becoming more Atlantic in character, with a higher productivity at the sea floor (*Cochrane et al., 2009*). The northern fish fauna currently has low biomass, but a shift towards more productive Atlantic water will likely result in an overall increase in benthic biomass to the north.

A northern shift in the penetration of Atlantic water will likely make the area more similar in faunal structure and ecosystem function to the southern parts of the Barents Sea. The Barents Sea today supports large commercial fisheries, and a potential climate-driven increase in their harvestable areas is of high social and economic interest. Under warming conditions, trophic interactions in the ecosystem could weaken as a result of increased diversity at each trophic level caused by range expansion of species found in warmer areas. Cod larvae are spread by currents from spawning grounds throughout the Barents Sea. As the waters around FJL warm, Atlantic cod and other boreal and sub-Arctic species will likely become more abundant. The impact of fisheries on the Arctic, which can be expected to increase, as industrial fisheries move into a warming Arctic following the invasion of boreal species. The lack of basic knowledge regarding fish biology and habitat interactions in the north, complicated by scaling issues and uncertainty in future climate projections limits our preparedness to meet the challenges of climate change in the Arctic with respect to fish and fisheries (*Reist et al., 2006*).

## Abbreviations

| | |
|---|---|
| **dp** | depth |
| **coll.** | collector |
| **I.** | Island |
| **lc** | head length |
| **spl** | sample |
| **st** | station |
| **sp** | specimen |
| **SL** | standard length (from tip of snout to bases of caudal-fin rays) |
| **TL** | total length |
| **DC** | drop camera observations |
| **MC** | materials collected |
| **VO** | visual observations |

*Fins*

| | |
|---|---|
| **A** | anal |
| **D** | dorsal |
| **C** | caudal |
| **P** | pectoral |
| **V** | ventral |

### Funding
Funding for this work was provided by National Geographic, Blancpain, Davidoff Cool Water, the Russian Arctic National Park, the Russian Geographical Society, and the State Nature Reserve—Franz Josef Land. The funders had no role in study design, data collection and analysis, decision to publish, or preparation of the manuscript.

### Grant Disclosures
The following grant information was disclosed by the authors:
National Geographic.
Blancpain.
Davidoff Cool Water.
The Russian Arctic National Park.
The Russian Geographical Society.
The State Nature Reserve—Franz Josef Land.

### Competing Interests
Natalia V. Chernova is an employee of the Zoological Institute of Russian Academy of Sciences. Alan M. Friedlander, Alan Turchik and Enric Sala are members of the National Geographic Society.

### Author Contributions
- Natalia V. Chernova, Alan M. Friedlander, Alan Turchik and Enric Sala conceived and designed the experiments, performed the experiments, analyzed the data, contributed reagents/materials/analysis tools, wrote the paper, prepared figures and/or tables, reviewed drafts of the paper.

### Animal Ethics
The following information was supplied relating to ethical approvals (i.e., approving body and any reference numbers):

Approval to conduct research on vertebrate animals was granted by the Russian Federation Ministry of Education and Science Approval Ref. No. 14-368 of 06.05.2013.

### Field Study Permissions
The following information was supplied relating to field study approvals (i.e., approving body and any reference numbers):

Approval to conduct field studies was granted by the Russian Federation Ministry of Education and Science Approval Ref. No. 14-368 of 06.05.2013 permission to the Russian Arctic National Park.

### Supplemental Information
Supplemental information for this article can be found online at http://dx.doi.org/10.7717/peerj.692#supplemental-information.

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
