# Peer review of "Franz Josef Land: extreme northern outpost for Arctic fishes"

_PeerJ, doi:10.7717/peerj.692_

## Round 0.1 · original submission · Minor Revisions

Both reviewers are, quite rightly, very positive about your paper and some helpful comments have been provided. I have a few minor observations:

94 - Tape-body, should that be tape-body? (i.e. lower case)
175 - check your usage of fish v fishes throughout the manuscript. Fishes are plural. Would you say several species of dog or several species of dogs? (I'd say dog)
210 - A 2 m (maybe?)
272 - Fish (plural for several specimens of the same species, fishes = several specimens of different species)
565 - facultative
648 - Until recently . . .

·

Basic reporting

The authors provide a fine contribution to baseline understandings of the ichthyofauna of Franz Josef Land Archipelao,which, until this study, have been poorly researched.

The manuscript is written in good English and is structured in a way which conforms to Peer J policies. There are however, some inconsistencies and some referencing which needs addressing:

Line 40: 'Mechlenburg' is spelt incorrectly, but is correctly spelled elsewhere and within the reference list as 'Mecklenburg'.

Line 82: Polar cod's scientific name has been given prior (line 74), but scientific names for twohorn sculpin and kelp snailfish have not been given. As this is the first instance of their mention, I would like to see their scientific names written at this point.

Line 158: 'weight.' I believe this is a typo and should read 'weighed'.

Line 203: Inconsistencies with date format. Given here as '13 August,' but dates are given elsewhere in the format of 13.08.2013 (line 409) and 17.08.2013 (line 415). Please ensure consistency and stick with only one of those two formats.

Line 210: 'One two m...' would read better, and would be consistent with later,similar size reports, as 'One 2m...', giving a numeric before the unit.

Line 214: To be consistent with the rest, please put the figure's caption below the figure.

Line 235: 'Knipowitch, 1901 et al.' I do not believe the 'et al' should be here.

Line 258: 'was' should read 'were.'

Line 273: '<0'. Although I, and I'm sure others, would assume the unit to be in degrees Celcius, please add the unit of temperature here.

Line 311: The pers. comm reference does not comply with Peer J - "...should be referred to as "pers. comm." followed by the relevant year."

Line 324: It might be more appropriate to write 'authority unknown' here, rather than a '?'.

Line 328: the species name 'grille' should be spelt 'grylle' as correctly written in line 331.

Line 413: 'was' should read 'were'.

Line 449: 'was' should read 'were'.

Line 494: 'was' should read 'were'.

Line 566: I believe 'major' should read 'majority'.

Line 637: As in your reference list, Welch et al's report was published in 1993, not 1992.


It also appears as though Barr (1994) is not listed in the text. Neither is Borkin IV(1994) , Chernova NV (1989) or Koltun VM (1964). Esipov VK (1931) exists, but in the reference list, is followed by a 'a', which is not needed. In the list, Fletcher.....2001 is written with an '&' between Hew and Davies, which is inconsistent with the rest of the referencing style. Peer J also asks for full Journal names, but this is not always given, for example with Chernova 2011: "J. Ichthyol."

Lastly, some references are not in alphabetic order, such as 'Savin,' which appears in the list before, for example, Stiansen. I would like the authors to carefully review their reference list in accordance with Peer J.

These errors/suggestions for improvements are minor and certainly do not detract from the value of the research.

Experimental design

For the purpose of this study, the methods are thorough and wide ranging, to gain the best possible data on ichthyofauna of FJL.

In the Materials and Methods section, I would like to see a few sentences on diel activities of FJL fishes. The authors may have had good reason not to sample at night (or perhaps they did, in which case, I don't feel it is emphasised) which may have influenced the results. Whatever the reason for sampling or not sampling during both day and night, I would like a few sentences to justify the author's choice, thus strengthening their methods. If the fishes in question do present differences in diel abundances, this may explain why abundances were low when scuba diving (line 612), if dives were conducted during the day, for example.

With line 146's reference to the frozen bait, I would like to see a little more information on bait type(s) and size/size ranges. It may be that difference sizes of bait is unimportant for deep-sea fish attraction, but I would like to see a sentence or two to highlight that this aspect was considered.

Ethical standards have been addressed and were given mention in the 'Sample design.'

Validity of the findings

Statistical analyses were not required for this research type.

Between diving, beach seines, plankton netting and use of cameras, I believe the authors captured a good representation of FJL fish life.

Line 648-654 mentions the apparently warming of the NE Barents Sea. In this changing environment, monitoring of ichthyofauna is important not only of commercial species, but also of those species which may be of prey items to commercial species, and this paper presents a good reference point for future studies.

Additional comments

With some minor suggestions/corrections as mentioned, I consider this paper an excellent baseline for FJL/NE Barents Sea ichthyofauna studies. The paper is informative, comprehensive and its photographs can also act as an identification guide for future FJL researchers.

·

Basic reporting

No comments

Experimental design

No comments

Validity of the findings

No comments

Additional comments

The authors deserve to be congratulated on their well-written, detailed report on the results of the 2013 expedition to FJL. The fish fauna of that remote area was poorly known, but this report represents a detailed assessment that includes: (1) history of earlier explorations, (2) relationship of the fauna to adjacent areas ( North Pacific, North Atlantic, Arctic Ocean, Barents Sea), and (3) estimates of endemism for the FJL and the Arctic. The results are important because they represent a maximum effort in terms of specimen collection and photography. Fishes were captured by use of beach seines, scuba diving, trawls, and plankton nets. Many of the excellent photographs were taken with newly-invented, deep water cameras. Included were detailed accounts of each species that described morphology, location of capture, and type of habitat. Also important were the ecological observations, especially the role of the polar cod, a cryopelagic species that provided a food resource for birds and mammals. Finally, this report comprises a well researched baseline by which future changes, such as those that may be caused by temperature increases, can be measured.

---

## Round 0.2 · accepted · Accept

Thank you for your amendments and corrections.